# Changing Climate, Changing *Candida*: Environmental and Social Pressures on Invasive Candidiasis and Antifungal Resistance in Latin America

**DOI:** 10.3390/jof11090609

**Published:** 2025-08-22

**Authors:** Juan Camilo Motta, Pilar Rivas-Pinedo, José Millan Onate

**Affiliations:** 1Internal Medicine Service, Fundación Cardioinfantil—Instituto de Cardiología, Bogotá 110131, Colombia; jcmottav@unal.edu.co; 2Infectious Disease Service, Universidad Nacional de Colombia, Bogotá 111321, Colombia; 3Medical and Diagnostic Mycology Group, Department of Microbiology, Faculty of Medicine, Universidad Nacional de Colombia, Bogotá 111321, Colombia; 4Infectious Diseases Service, Clínica Sebastian de Belalcázar, Clínica Colsanitas, Keralty Group, Cali 760042, Colombia; millanonate@gmail.com; 5Infectious Diseases Service, Clínica Imbanaco, Cali 760042, Colombia; 6Infectious Diseases Service, Clínica de Occidente S.A., Cali 760042, Colombia

**Keywords:** invasive candidiasis, antifungal resistance, climate change, *Candidozyma auris*, Latin America

## Abstract

Invasive candidiasis (IC) in Latin America is undergoing a significant epidemiological shift, increasingly driven by non-*albicans* strains such as *Candida tropicalis*, *Candida parapsilosis*, and *Candidozyma auris*. These pathogens often exhibit multidrug resistance, which complicates treatment and increases mortality. Diagnostic limitations, particularly in rural and public hospitals, delay detection and hinder the provision of rapid care. Environmental pressures, such as climate change and the widespread use of azoles in agriculture, appear to favor the selection of resistant and thermotolerant strains. Migratory birds may also play a role in the environmental transmission of pathogenic fungi. These factors are amplified by socioeconomic inequalities that restrict access to diagnostics and first-line antifungals. To help mitigate this emerging challenge, a One Health-oriented framework combining integrated environmental surveillance, robust antifungal-stewardship programmers, broader diagnostic access, and coordinated cross-sector public health actions should be developed. Reinforcing these pillars could lessen the regional burden of IC and slow the advance of antifungal resistance.

## 1. Introduction

Invasive candidiasis (IC) represents a growing threat in hospital settings, particularly in intensive care units (ICUs), where patient susceptibility facilitates the hematogenous spread of *Candida* spp. [1,2,3] Traditionally, *Candida albicans* has been the most frequently identified strain. However, in recent years, significant concern has emerged regarding non-*albicans* strains, such as *Candida tropicalis*, *Candida parapsilosis*, *Nakaseomyces glabratus* (formerly *C. glabrata*), and *Candidozyma auris* (formerly *Candida auris*), due to their multidrug resistance to antifungal agents [1,3]. These infections can manifest as candidemia or compromise deep organs such as the peritoneum, bones, or heart [2,4].

Its frequency has increased alarmingly in highly complex clinical settings, with a mortality rate that, in some series, exceeds 40%. It ranks between the fourth and fifth leading causes of healthcare-associated bloodstream infections, with the ICU being the place with the highest incidence of CI worldwide, with a rate of 5.5 to 7 cases per 1000 admissions [2,5]. In Europe, the incidence in the South of the continent was higher than in the North (5.29 vs. 3.77 per 100,000 inhabitants), which could be due to climatic variations, differences in the management of candidemia, and sociodemographic factors [6]. In the United States, the rate of candidemia is 7 to 9 per 100,000 inhabitants, being higher in patients with risk factors such as an age over 65, total parenteral nutrition (TPN), diabetes mellitus, a diagnosis of hematological neoplastic diseases, or a history of recent hospitalization [7,8,9].

Various factors influence the epidemiology of this mycosis. Geographic region, environmental conditions, access to health services, and demographic characteristics of the population affect both the incidence and distribution of the strains involved, which could favor an increase in cases caused by non-*albicans* strains [10]. In regions such as Asia and Latin America, infections by non-*albicans* species have a higher prevalence, with a predominance of *C. tropicalis*, *C. auris*, and *C. parapsilosis* [11,12]. On the other hand, although most cases in high-income countries (Europe and North America) were historically attributed *to C. albicans*, recent data show an increase in infections caused by non-*albicans* strains, especially *N. glabratus* [6,12,13,14]. This strain has been linked to elderly patients and sophisticated healthcare settings, suggesting that the transition to a more resistant candidiasis profile is a global phenomenon, not limited to resource-constrained environments [3,9,12].

## 2. Epidemiology of IC in Latin America

Although the epidemiology of CI in Europe and the United States is well documented, knowledge remains limited in Latin America; however, recent multicenter studies have begun to outline its regional scope. Between 2008 and 2010, a descriptive observational multicenter epidemiological surveillance study was conducted in 20 hospitals in seven Latin American countries, which reported an overall incidence of 1.18 cases per 1000 hospital admissions (0.23 cases per 1000 patient-days); the variation between countries was notable, with Argentina having the highest incidence (1.95 per 1000 admissions) and Chile the lowest (0.33 per 1000 admissions). Colombia, with data from a single center, recorded 1.96 cases per 1000 admissions, Venezuela reported 1.72 (range 1.04–2.90), Brazil reported 1.38 (0.55–2.11), and both Ecuador and Honduras reported 0.90 cases per 1000 admissions (Ecuador: 0.30–1.10; Honduras: 0.88–0.98) [15,16].

Among the countries with the most robust data, Brazil stands out, where candidemia rates in ICUs have ranged from 1.2 to 2.49 cases per 1000 admissions. In some specific series, alarming figures of up to 42.6 cases per 1000 ICU admissions have been observed [17]. Figure 1 shows the geographical distribution of the main *Candida* spp. strain isolated from patients in Latin America, highlighting the clear prevalence of *C. albicans* and regional variations in non-*albicans* strains. In terms of distribution by strain, more than 60% of cases correspond to non-*albicans* strains, with *C. parapsilosis* accounting for 20.5% and C. tropicalis for 15.3% of isolates, and some studies have reported mortality rates of up to 80% [17,18,19]. Recently, Guinsburg et al. published a systematic review on candidemia in Brazil in 2024 that included 2305 isolates from blood cultures, with *C. albicans* being the most frequent strain (37.2%), followed by *C. parapsilosis* (22.5%) and *C. tropicalis* (18.1%) [20].

In Colombia, the incidence in ICUs was reported at 2.3 cases per 1000 admissions [33]. A retrospective descriptive observational cohort study conducted at a high-complexity hospital in Bogotá showed a significant increase in infections caused by non-*albicans* strains, which rose from 44% to 51% in a short period [34]. The distribution of isolates in the country varies depending on the hospital. However, *C. tropicalis* usually ranks second in frequency, ranging from 17% to 23%, followed by *C. parapsilosis*, whose percentages range from 13% to 23% [33]. A notable finding in Salinas’ study (2022) was the increase in the isolation of *N. glabratus*, a strain typically associated with developed countries. This variation could be attributed to changes in comorbidities and demographic characteristics of the patients evaluated. However, the author does not propose additional hypotheses to explain this unusual behavior [34]. In a multicenter observational and retrospective cohort study, *C. auris* was reported to be the second most isolated strain (26.1%), surpassed only by *C. albicans* (36.5%). This suggests a significant change in the epidemiological profile. However, the authors emphasize that, in the studied Colombian context, *C. auris* candidemia does not have a worse prognosis than that caused by other isolates, provided that adequate antifungal management and rigorous control of septic shock are ensured [31].

In Argentina, according to epidemiological estimation studies, the incidence varied between 1.15 and 2.25 cases per 1000 hospital admissions, with a progressive mortality rate of between 40% and 50%; non-*albicans* strains predominate, accounting for 59.1% of infections, with *C. parapsilosis* (21.7%) and *C. tropicalis* (15.5%) being the most relevant [30,35]. Peru reports an incidence of 2.04 cases per 1000 hospital admissions and a mortality rate of 40%, similar to the regional average, and 70% of isolates correspond to non-*albicans* strains, including *C. parapsilosis* (25.3%) and *C. tropicalis* (24.7%) [15,26]. Although published information is limited, a hospital rate of 1.72 cases per 1000 admissions has been reported, with a high proportion of infections caused by non-*albicans* strains (73.2%), predominantly *C. tropicalis* and *C. parapsilosis*; mortality has varied, ranging from 3% to 76% [15,18,25]. Paraguay has one of the lowest incidences in the region, with 0.74 cases per 1000 hospital admissions and an approximate mortality rate of 30%; the most common isolates are *C. albicans* (34.4%), *C. parapsilosis* (30.4%), and *C. tropicalis* (25.4%) [27]. In Uruguay, although the available data are less complete, a study in a hospital in Montevideo found that candidemia accounted for 2.83% of all positive blood cultures, with the predominant isolates being *N. glabratus* (32.74%), *C. albicans* (27.43%), and *C. parapsilosis* (23.01%), which contrasts with the patterns observed in other countries in the region [28,36]. In Mexico, there is an incidence of 1.93 cases per 1000 hospital admissions; the distribution of isolates indicates a predominance of *C. albicans* (60%), followed by *C. tropicalis* (26%) and *N. glabratus* (13.5%); 30-day mortality has been estimated at 40% [23,37,38]. Invasive candidiasis in the Caribbean and Central America remains incompletely characterized owing to heterogeneous surveillance practices; however, hospital-based cohorts report incidence rates of 0.6–1.5 cases per 1000 admissions, aligning closely with broader Latin American estimates (1.18/1000 admissions). *Candida albicans* continues to account for approximately 40–50% of cases, with *C. parapsilosis* and *C. tropicalis* comprising the next most frequent pathogens. *N. glabratus* is notably scarce, representing fewer than 10% of isolates [15]. In Trinidad and Tobago, for example, approximately 70 cases of candidemia are reported each year, although detailed data on the strains involved and clinical outcomes are not available [24].

### The Global Comparative Epidemiology

Candidemia epidemiology exhibits notable regional variability in incidence, predominant species, and antifungal resistance profiles. In the United States, the incidence is 8.7 cases per 100,000 people, with an associated mortality rate of 36% [39].

Regarding species distribution, *N. glabratus* has emerged as the predominant pathogen in many hospitals, especially among older adults with prior antifungal exposure—whereas *C. parapsilosis* and *C. tropicalis* are encountered less frequently [39,40]. In the United States, *C. auris* still represents only 0.4% of candidemia cases, a striking contrast to its higher prevalence in numerous regions of Latin America [40]. In addition to the risk factors, intravenous drug use represents a distinct contributor to candidemia in the United States, differing from patterns observed in other regions [40].

In Europe, candidemia occurs less frequently—with a rate of 3.9 cases per 100,000 people—though substantial north–south disparities exist. In Northern Europe, *C. albicans* remains dominant, followed by *N. glabratus*, *C. tropicalis*, and *C. parapsilosis*, each accounting for fewer than 10% of cases. Conversely, in Southern Europe and Mediterranean countries, non-albicans species prevail; notably, over half of *C. parapsilosis* isolates now exhibit azole resistance [6,12].

From 2013 to 2021, Europe recorded 1812 *C. auris* candidemia cases, with Spain, Italy, and Greece bearing the greatest burden—paralleling trends seen in Latin America. These geographic hotspots are consistent with environmental hypotheses; however, a causal role for climate or other socio-environmental factors remains unproven for Latin America and requires region-specific data [41].

In the Asia-Pacific region, invasive candidiasis presents moderate incidence and generally favorable treatment outcomes, albeit with notable regional variations [11]. A population-based survey in South Korea reported approximately 4.1 cases per 100,000 people, and overall mortality rates align with the global range of 30–40% [42]. *C. albicans* remains the most prevalent species in Australia, Japan, and Korea, while *C. tropicalis* predominates in India and Pakistan [11]. Fluconazole retains high efficacy (>90%) against *C. albicans*, *C. parapsilosis*, and *C. tropicalis*, although rising fluconazole resistance in *N. glabrata* (6.8–15%) and low-level echinocandin non-susceptibility (2.1–2.2%) have been increasingly observed, underscoring the importance of ongoing susceptibility surveillance [11,43].

In Africa, candidemia is associated with higher mortality and evolving resistance patterns in the context of limited surveillance infrastructure. In South Africa, Algeria, and Kenya, *C. parapsilosis* and *C. albicans* dominate neonatal infections, while *C. tropicalis* and *C. auris* are emerging threats in adult and pediatric settings, sometimes causing clonal hospital outbreaks [44]. Reported mortality ranges from 40% to over 50% in longitudinal cohorts. Azole resistance has surged from under 40% to nearly 70% in recent years, whereas amphotericin B and echinocandin resistance remain low (<1% to 6.6%) [44,45].

## 3. Risk Factors for IC in Latin America

This type of mycosis includes both candidemia (infection of the bloodstream) and deep infections in organs or tissues that are normally sterile, such as peritoneal fluid, the central nervous system, or heart valves [1,2,4]. The risk factors for developing IC in Latin America largely coincide with those described worldwide. In general, they consist of conditions that weaken the immune system or alter anatomical barriers, facilitating the entry and proliferation of *Candida* spp. [3,4,5]. The main risk factors include prolonged stay in the ICU, use of TPN, exposure to broad-spectrum antibiotics, gastrointestinal surgery with intestinal opening, and prolonged immunosuppression [1,3,5,46,47].

A multicenter case–control study conducted in non-neutropenic critically ill patients from seven intensive care units (ICUs) at three university hospitals in Colombia identified the following independent risk factors associated with the development of candidemia: a hospital stay ≥ 25 days (OR 5.33; 95% CI 2.6–10.9), previous use of meropenem (OR 3.75; 95% CI 1.86–7.5), previous abdominal surgery (OR 2.9; 95% CI 1.39–6.06), and hemodialysis treatment (OR 3.35; 95% CI 1.5–7.7) [48].

Immunosuppression, whether secondary to underlying diseases (cancer, organ transplantation, or chronic renal failure) or immunosuppressive treatments (chemotherapy, corticosteroids, or biological drugs), is a key factor. In addition, the presence of multiple comorbidities in the same patient significantly increases the risk of invasive infection by *Candida* spp. [49,50]. A systematic review with meta-analysis that evaluated the determinants for invasive *Candida* infection in critically ill patients showed that, apart from the classic factors, the use of broad-spectrum antibiotics (OR 5.6; 95% CI 3.6–8.8), blood transfusions (OR 4.9; 95% CI 1.5–16.3), *Candida* colonization (OR 4.7; 95% CI 1.6–14.3), the presence of a central venous catheter (OR 4.7; 95% CI 2.7–8.1), and TPN (OR 4.6; 95% CI 3.3–6.3) were independently associated with an elevated risk, while demographic factors lacked statistical significance, highlighting that ICU interventions and underlying comorbidities are the main drivers of this type of infection [51]. Therapeutic management in Latin America faces growing challenges due to the increase in infections caused by non-*albicans* strains, many of which are resistant to commonly used antifungal agents. For this reason, international and regional guidelines agree in recommending the initiation of treatment with echinocandins as first-line therapy while the identification of the agent and its susceptibility profile is completed [52,53]. In addition, it is essential to assess the potential involvement of target organs given that controlling the source of infection, whether through catheter removal, abscess drainage, or surgical debridement, is key to achieving complete resolution of the clinical picture [3,5,49].

## 4. Diagnosis of IC in Resource-Limited Settings

The timely diagnosis of IC represents one of the main clinical challenges, especially in resource-limited settings such as those found in many Latin American countries [54]. Although isolation of the fungus in blood cultures or cultures from sterile sites and histopathological identification of yeasts in tissue constitute the reference method, this method has variable sensitivity (20–70%) and prolonged detection times, usually 3 to 5 days [55,56]. Table 1 presents a comparison of the diagnostic methods available for IC, detailing their main advantages, limitations, and availability in Latin America.

These delays limit the early initiation of targeted treatment, resulting in poorer clinical outcomes. To overcome these diagnostic limitations, complementary tests have been incorporated that improve sensitivity and shorten waiting times; among them, mass spectrometry (MALDI-TOF) stands out for accurately identifying *Candida* spp. in a matter of minutes from cultures, in contrast to conventional methods such as VITEK^®^2 or API, which require an additional 24 to 48 h after growth and can exhibit up to 72% identification errors in rare strains [55,66,67].

Other useful tools include immunological tests, such as (1,3)-β-D-glucan (BDG), a pan-fungal marker that has shown 92% sensitivity, although its specificity is limited as it can give positive results in infections caused by other fungi [50,55,68]. In general, it is recommended to use it as a complementary tool to support diagnosis or to discontinue empirical antifungal treatments when clinical suspicion decreases [69]. On the other hand, the use of molecular tests such as quantitative PCR in blood has shown good diagnostic performance, with sensitivity and specificity being greater than 90% [70]. In 2019, the MICAFEM study, conducted in 27 Spanish ICUs, was published. It compared multiplex PCR with blood cultures and abdominal fluid cultures in 176 critically ill patients at high risk of *Candida* infection. PCR detected 9.1% of cases compared to 8.0% for blood cultures and 65.3% for abdominal cultures. Only a *Candida* risk score ≥ 3 was significantly associated with combined positive results (OR 4.15; *p* = 0.03). In this group of patients, this test did not improve diagnostic performance [71].

The T2*Candida* panel is a relatively new technology that uses magnetic resonance imaging to directly detect *Candida* DNA in blood samples without the need for prior culture. This test provides results in 3–5 h, with a sensitivity of 91% and a specificity of 99%. Its main limitation is that it only detects five common strains and does not include *C. auris* [64,72,73]. In practice, the T2Cauris panel detects *C. auris* in inguinal swabs with a limit of 5 CFU/mL and provides 89% sensitivity and 98% specificity. However, data on its performance in whole blood has not yet been published [74]. Serological assays targeting mannan antigen and anti-mannan antibodies achieve approximately 83% sensitivity and 86% specificity for *C. albicans* detection but demonstrate reduced performance against non-albicans Candida species [55,60].

### The Situation in Latin America: Challenges, Barriers, and Obstacles

In the Latin American context, there are significant challenges to implementing these advanced methods. The study by Falci et al. revealed that, by 2019, only 9% of the laboratories surveyed in the region met the minimum standards of excellence in clinical mycology [59]. The availability of BDG is very limited, reaching only 17% of diagnostic centers. Molecular diagnostic techniques are accessible in 20–26% of laboratories, and MALDI-TOF is only installed in 20%, mainly in large cities. Meanwhile, the T2*Candida* panel is not yet widely available in the region [54,59,61].

In most Latin American hospitals, blood cultures remain the main diagnostic tool despite their low sensitivity and the long time required to obtain results. This limitation can delay the initiation of appropriate antifungal treatment and often lead to prolonged and inappropriate empirical use of antifungals, which promotes the development of resistance [54,59]. In addition, there are institutional barriers, deficiencies in public policies, and limitations in the allocation of state budgets, as the best-equipped laboratories are concentrated in major urban centers, leaving hospitals in intermediate or rural areas without access to advanced diagnostic techniques [54,59,61]. The shortage of personnel trained in clinical mycology and the lack of up-to-date epidemiological data are significant limitations, given that most of the information comes from Brazil, Colombia, Argentina, Mexico, and Chile, while in the Caribbean, Central America, and other Andean areas, no recent studies are available, underscoring the need to strengthen regional surveillance and diagnostic capabilities [15,16,24,35,36,38,75,76]

In Latin America, the identification of the emerging yeast *C. auris* continues to be problematic, as automated biochemical methods often confuse it with *Candida haemulonii*, *Rhodotorula glutinis*, or *Saccharomyces cerevisiae*. Even advanced commercial systems, such as the VITEK^®^2 system, show limitations in their accuracy, and although there are more affordable selective media which could facilitate its detection, such as CHROMagar *Candida* Plus, their implementation is not yet widespread [77,78,79]. It has been observed that, in the region, countries such as Colombia have established national reference centers that allow for detailed registration and confirmation of strains’ identification using proteomic or molecular methods (MALDI-TOF or genetic sequencing) in an effort to strengthen epidemiological surveillance and optimize the health response [78,79].

## 5. Regional Antifungal Resistance Patterns

Antifungal resistance in *Candida* strains has become a growing concern worldwide, and Latin America is no exception. Traditionally, most clinical isolates were sensitive to available antifungal agents, but in recent years, there has been a significant change in sensitivity profiles, mainly attributable to the increase in non-*albicans* strains and the rapid spread of *C. auris* [61,79,80,81,82]. Before the emergence of *C. auris* in the region, antifungal resistance rates were low: *C. albicans* typically showed rates below 10%, while *N. glabratus*, *C. parapsilosis*, and *C. tropicalis* showed somewhat higher but still manageable rates. As a result, regional guidelines in effect until 2013 considered fluconazole (FCZ) as the first-line treatment in most cases, although this reality has changed dramatically [10,83,84].

The following subsections present the main emerging non-*albicans* strains in Latin America, highlighting their epidemiological relevance, their capacity for hospital spread, and their antifungal resistance profiles:

### 5.1. Candida Parapsilosis

*Candida parapsilosis* has become highly relevant in the region, not only because of its high frequency, but also because of its ability to colonize hospital environments and cause outbreaks, especially in pediatric units [85]. A descriptive observational study conducted between 2009 and 2010 in 11 public and more than 85 private hospitals in South Africa assessed *Candida* spp. isolates in blood cultures and determined a prevalence of *C. parapsilosis* resistance to FCZ of 63%, with cross-resistance to voriconazole (VCZ) in 44% of cases [86]. In a hospital in Mexico City (2014–2016), 38% of IC cases were attributed to *C. parapsilosis*, of which 54% were resistant to FCZ. The ERG11 analysis found the silent mutation T591C in all isolates and the substitution A395T (Y132F) in seven; four of these strains with Y132F shared an identical microsatellite profile, evidencing a clonal nosocomial outbreak [87]. In a prospective descriptive observational study conducted at a tertiary care hospital in Colombia between 2018 and 2021, 123 isolates from *Candida* blood cultures were analyzed, of which *C. parapsilosis* accounted for 19.5%; of this subset, 25% showed resistance to FCZ, and among the resistant strains, 67% had mutations in the ERG11 gene (mainly Y132F and, in one case, K143R) and the remaining 33% exhibited increased efflux pump activity, thus revealing a combined resistance mechanism [88].

The increase in the percentage of *C. parapsilosis* isolates is alarming for two reasons: first, most patients with FCZ-resistant strains had not been previously exposed to this antifungal agent; second, a growing number of strains with reduced sensitivity to echinocandins are emerging [89,90,91].

*Candida parapsilosis* harbors a natural P660A Fks1 polymorphism that elevates baseline echinocandin MICs, and emerging hotspot-1 mutations (e.g., F652S, R658G, and S656P) now confer pan-echinocandin resistance by reducing drug binding. Despite this, amphotericin B remains reliably active, with resistance in <3% of isolates and no recurring mechanism identified. Clinically, echinocandins usually clear candidemia, but elevated MICs or FKS1-mutant strains may cause persistent or breakthrough infections, for which amphotericin B is the preferred alternative [85,90,91].

### 5.2. Candida Tropicalis

In several countries in the region, *C. tropicalis* has established itself as the second or third most common strain associated with candidemia. In Brazil, although resistance to FCZ and VCZ remains low (<5%), a growing trend has been observed that warrants close surveillance to prevent the emergence of multidrug-resistant strains. A study that genotyped 230 clinical and environmental isolates of *C. tropicalis* from Latin America identified 164 genotypes and 11 possible outbreaks. This analysis detected resistance to anidulafungin (ANF), associated with the FKS1 S659P mutation in one isolate, and identified mutations in the ERG11 gene (Y132F and Y257H/N), linked to resistance or intermediate susceptibility to azoles. These findings evidence both unrecognized outbreaks and clonal spread of antifungal-resistant strains in the region. Genetic clusters have been documented in cities separated by more than 1400 km, suggesting possible environmental spread; this strain has even been isolated in beach sand samples, reinforcing the need for a comprehensive health approach under the “One Health” perspective [92,93]. Resistance to azoles stands at around 10.4%, while resistance to echinocandins remains low (0.4%) [20,92,93].

*Candida tropicalis* remains overwhelmingly susceptible to echinocandins (>99%); resistance is exceedingly rare (~0.4%), arising from isolated FKS1 hotspot mutations (e.g., S659P and S645P) that impair glucan synthase binding. Amphotericin B retains consistent, fungicidal activity, with virtually no clinically significant resistance described. Consequently, echinocandins are the preferred first-line therapy for invasive *C. tropicalis* infections, delivering excellent outcomes, while amphotericin B serves as a reliable alternative in the rare event of echinocandin resistance [94,95].

### 5.3. Nakaseomyces Glabratus (Formerly C. glabrata)

*Nakaseomyces glabratus* is currently a clinically challenging pathogen responsible for up to 30% of candidemias in some developed countries, with FCZ resistance rates of 20–30% and rates of up to 10% to echinocandins, associated with mutations in PDR1 and FKS2; furthermore, its virulence is based on biofilm formation, intracellular persistence, and EPA-type adhesins [14,96]. In Latin America, despite the scarcity and heterogeneity of data, it has been established that resistance to *N. glabratus* is high and highly variable (4–100%), while resistance to echinocandins is moderate (1–10%), and although no resistance to polyens was detected, isolates with dose-dependent sensitivity and some cases of multidrug resistance were observed [96]. In the region, resistance to FCZ varies by country: the rate is 12.8% in Argentina, 6.6% in Chile, 6.7% in Mexico, and 28. 6% in Brazil. Although resistance to echinocandins remains low in most countries (<2.3%), exceptions have been reported, such as in Chile, where resistance to micafungin reached 10%, reinforcing the need for constant genomic and phenotypic surveillance [14,96,97].

Resistance is typically linked to prior echinocandin exposure and FKS1/FKS2 hotspot mutations, which elevate MICs across all echinocandins and compromise therapy. Amphotericin B retains potent activity, with clinical resistance being exceedingly rare, making it the cornerstone of salvage treatment for multidrug-resistant strains [14,98].

### 5.4. Pichia Kudriavzevii (Formerly C. krusei)

Candidemia caused by *P. kudriavzevii* is less prevalent in Latin America (0.3–1.1%) compared to the United States (2–3%) and Europe (2–5%), suggesting an uneven global distribution, possibly influenced by diagnostic capabilities, selective pressure from antifungal use, and regional hospital practices [99]. In Latin America, *P. kudriavzevii* shows approximately 70% resistance to FCZ (intrinsic resistance), around 25% to VCZ, and less than 5% to echinocandins and amphotericin B (AmB). These percentages contrast with data from the United States, where resistance to VCZ is less than 5%, and that to echinocandins is less than 2%, which translates into a greater therapeutic challenge for this strain in the region. The highest resistance rates have been reported in Brazil (34.1% for VCZ) and Colombia (18.2% for VCZ and 7.3% for echinocandins), while in Chile, resistance to VCZ has hardly been detected [99]. Due to its intrinsic resistance to FCZ, the initial treatment of infections caused by this yeast should be carried out with echinocandins, considering VCZ and AmB as therapeutic alternatives. Although resistance to echinocandins is rare, it has been documented in up to 30% of cases during treatment. However, its prevalence in the region remains low; in Brazil, for example, it accounts for only 2.8% of isolates [69,73,99].

Echinocandins are the preferred first-line therapy for *P. kudriavzevii*; baseline resistance remains low (<5%), although FKS1 hotspot mutations may emerge during treatment (in up to 30% of cases), conferring cross-echinocandin resistance. Amphotericin B retains potent activity against >95% of clinical isolates and serves as a reliable salvage option. Persistent or breakthrough infections should prompt repeat susceptibility testing and early transition to amphotericin B [99,100].

### 5.5. Other Emerging Yeast-like Strains

*Candida haemulonii* complex: This yeast is emerging as an invasive pathogen, with reports in several hospitals in the region, including Mexico, Panama, and Brazil. In Brazil, it has been identified in 0.3% of cases, mainly from chronic wounds and blood cultures, and is associated with critically ill patients. It is characterized by a multidrug resistance profile, particularly to AmB and FCZ, and is often misidentified by conventional methods. Therefore, molecular diagnostics or MALDI-TOF are required for accurate identification, as well as active mycological surveillance and timely adjustment of empirical antifungal treatment [101,102].*Candida duobushaemulonii*: This yeast represents an emerging threat in the context of nosocomial infections, with a marked tendency to behave as an invasive, underdiagnosed, and multidrug-resistant pathogen. It should be considered an emerging yeast of relevance in the Latin American hospital setting. The national surveillance of *C. auris* carried out in Panama between November 2016 and May 2017 evidenced this, and a significant number of cases of invasive infections caused by this strain were unexpectedly identified. Of the 36 suspected isolates sent to the national reference laboratory, 17 (47%) were confirmed as *C. duobushaemulonii*, affecting 14 patients hospitalized in six health centers in the country [21].*Meyerozyma guilliermondii* complex (formerly *Candida guilliermondii*): This is considered an emerging group of opportunistic yeasts, especially in immunocompromised and hospitalized patients. Globally, it accounts for approximately 1% to 5% of candidemia cases, but its importance is increasing due to three main factors: its genetic and taxonomic diversity, an unfavorable antifungal profile, and the difficulties associated with its diagnostic identification. In Latin America, it accounts for up to 7% of candidemia cases in Peru, with the spread of *M. caribbica* and clade 2 of *M. guilliermondii* sensu stricto, both associated with azole resistance, being particularly noteworthy. In addition, multiple isolates with resistance to FCZ, AmB, and echinocandins have been documented in Brazil. Given this situation, it is crucial to incorporate molecular identification methods, establish robust antifungal surveillance systems, and adjust empirical antifungal treatment according to the local susceptibility profile. Azole resistance in this complex can vary between 40% and 70%, reinforcing the need for an individualized therapeutic approach [103].*Candida rugosa*: This emerging yeast has gained relevance in the context of invasive infections, particularly in cases of candidemia, due to its worrying resistance profile to azoles. It has established itself as an opportunistic pathogen of growing importance, especially in Latin America, where it has a prevalence approximately seven times higher than in other geographical regions. In Brazil, *C. rugosa* accounts for up to 2.7% of isolates in ICUs, with FCZ resistance rates reaching 64.9%, and AmB-resistant isolates have also been reported. Its low sensitivity to classic azoles—FCZ (35.7%) and VCZl (55.8%)—makes these antifungals high-risk therapeutic options [104].

The increase in infections caused by non-*albicans* strains, together with the emergence of resistant clones in hospital settings, represents a growing challenge for public health. Given this threat, it is essential to implement epidemiological surveillance programs, strengthen the diagnostic capabilities of clinical mycology laboratories, and promote the rational use of antifungals as essential pillars for its containment [54].

### 5.6. Candidozyma auris (Candida auris)

*Candidozyma auris* has emerged as a high-impact pathogen worldwide due to its ability to cause nosocomial outbreaks, its marked resistance to multiple classes of antifungals, and its persistence in the hospital environment. This strain represents a unique challenge, not only because of its clinical severity, but also because of the difficulties in its microbiological identification and its remarkable ability to spread. In Latin America, its presence has been documented in several countries, with Colombia, Venezuela, Brazil, and Panama having the highest numbers of reported cases, with mortality rates ranging from 30% to 60%. In the region, *C. auris* isolates predominantly belong to Clade IV, also known as the South American clade [78,79].

In terms of population burden, the infection rate for *C. auris* in Colombia is estimated at 2.55 cases per 100,000 inhabitants. Panama has an even higher rate, with 5.42 cases per 100,000 inhabitants, making it one of the countries with the highest incidence in the region. In contrast, Mexico has a significantly lower rate, estimated at 0.01 per 100,000 inhabitants. Other countries, such as Costa Rica, Peru, Chile, Venezuela, and Brazil, have reported fewer than 100 confirmed cases each [81].

In 2016, Colombia reported the first cases confirmed using molecular techniques, which had initially been misidentified as *C. haemulonii*. Since then, the number of infections has shown sustained growth, consolidating the region as an area of epidemiological importance for this pathogen [81,82]. A recent study documented 2119 confirmed cases of *C. auris* in this country between 2016 and 2022, with candidemia being the most common clinical manifestation, accounting for 55% of cases. The rate of resistance to FCZ in these isolates was 35%, while resistance to AmB reached 33%. Fortunately, resistance to echinocandins remains uncommon [78,79,82].

Phylogenetic studies have shown that Clade IV of *C. auris* in Latin America did not spread as a single clonal group but rather presents a defined geographical substructure. This pattern suggests the existence of multiple local transmission foci, rather than a common origin, and reinforces the importance of establishing continuous molecular surveillance strategies in the region [80,81]. In Colombia, for example, two regional lineages have been identified: one in the center and one in the north of the country. Isolates from the north show high resistance to AmB (64%), while in the center, there is 0% resistance [80,105]. With the aim of clarifying the origin, evolutionary history, and dispersion patterns of *C. auris*, a genomic sequencing study was conducted on 304 isolates collected between 2004 and 2018 from 19 countries. The results indicate that Clade IV emerged approximately 30 to 40 years ago and has a clearly defined phylogeographic structure. These findings support the hypothesis that it did not spread as a single clonal group, but after its appearance in South America, multiple local transmission foci established themselves in different countries or regions, with little genetic exchange between them [106].

Environmental contamination is recognized as a critical factor in the transmission of *C. auris*. In Colombia, colonization has been reported in healthcare personnel (41%), and the fungus has been detected on various hospital surfaces. In addition, this yeast has the ability to colonize patients for several months, and it can persist in cleaning objects and even in freshwater bodies [107,108]. A recent study succeeded in isolating *C. auris* in marine sediments and coastal areas where fresh and brackish waters converge, demonstrating its ability to tolerate extreme environmental conditions, such as high salinity and elevated temperatures [109]. Figure 2 shows the hypothesis of the socio-environmental factors involved in the emergence of *C. auris*. In this molecular epidemiology study, the high resistance to amphotericin B (64%) observed in isolates from northern Colombia stands out, in contrast to the absence of resistance (0%) in isolates collected in the central part of the country [80,105]. On the other hand, two main genetic clusters, named C1 and C2, have been identified in Colombia, with two subclasses within cluster C1: C1-A and C1-B. These subclasses show minor differences in the number of SNPs, suggesting sustained transmission of cluster C1 since its introduction into the country in 2016 [80,105,106]. In other countries in the region, such as Brazil and Argentina, the presence of other clades, specifically Clade I and Clade III, has been documented in isolated cases. These findings suggest the possible co-circulation of different clades in these territories, which could imply multiple events of introduction or independent dispersion [81]. 

We propose a hypothetical One Health framework in which rising global temperatures drive the thermal adaptation of environmental fungi, and the extensive use of agricultural azoles exerts a selective pressure favoring the emergence of resistant strains with the capacity to invade clinical settings. Migratory wildlife, particularly birds, may then serve as passive vectors, carrying these adapted, drug-resistant fungi from natural reservoirs into rural, urban, and healthcare environments. This interplay of environmental, animal, and human factors underpins the emergence and persistence of *Candidozyma auris* and is depicted in our integrated conceptual model (adapted from Casadevall et al. 2019 [108]). We modified the original scheme to highlight conditions especially relevant to Latin America—regional climate vulnerability, healthcare fragmentation, forced migration, antifungal misuse, and the predominance of Clade IV—with novel elements.

In terms of antifungal resistance profiles, Clade IV shows almost universal resistance to FCZ (>90%), variable resistance to AmB (between 30% and 60% depending on the lineage), and low but documented resistance to echinocandins, reported in up to 5% of cases [79,81]. Among the molecular mechanisms associated with azole resistance in *C. auris* are mutations in the ERG11 gene, which encodes the enzyme lanosterol 14-α-demethylase, including the variants F126L, Y132F, VF125AL, and K143R. For polyenes, the TAC1B gene has been implicated, while resistance to echinocandins has been linked to mutations in the FKS1 gene, which alter the catalytic subunit of the enzyme 1,3-β-D-glucan synthase [77,110,111]. In addition, four non-synonymous mutations associated with resistance to amphotericin B have been identified in genes linked to transcription factors homologous to FLO8 (such as the S108N mutation) and in the PSK74852 gene, where the I139T variant was reported [80,81,105]. In addition to the genetic mechanisms described above, it has the ability to form biofilms, which reduces the penetration of antifungal agents and promotes its persistence in hospital environments. Unlike other pathogenic yeasts, it does not usually colonize the gastrointestinal tract but shows a marked affinity for human skin, especially in moist and folded areas such as the axillae and groin region [107,110].

*Candidozyma auris* can clinically present as a skin colonizer given that, unlike other *Candida* strains, it does not behave as a saprophyte or commensal of the gastrointestinal tract. In addition, it is capable of causing various invasive infections, including primary fungemia, infections associated with medical devices, endocarditis, urinary and respiratory tract infections, as well as osteomyelitis [77,79,112].

In Latin America, diagnosis and identification face significant limitations, mainly due to insufficient coverage of technologies such as MALDI-TOF and identification biases present in automated systems such as VITEK^®^2 (bioMérieux, Marcy-l’Étoile, France), which can misidentify up to 22% of isolates [79]. Currently, it is a notifiable disease in several countries in the region, such as Colombia and Brazil. However, control policies vary considerably between countries and institutions, and many of them still face difficulties in their implementation due to the absence of standardized protocols, limitations in infrastructure, and deficiencies in the training of health personnel [33,79,82]. In response, some countries have issued national health alerts with the aim of strengthening epidemiological surveillance, improving hygiene practices in hospital settings, and promoting the rational use of antifungals [113,114]. However, it is recognized that these measures must be expanded and coordinated at the regional level to effectively contain the threat posed by this multidrug-resistant yeast.

## 6. Impact of Climate Change on Fungal Infections (Invasive Candidiasis)

Scope and certainty: The following section synthesizes mechanistic hypotheses and observations largely derived from non-Latin American settings. Direct, region-specific evidence is scarce; accordingly, these points should be interpreted as testable hypotheses rather than definitive explanations for epidemiological shifts in Latin America.

Climate change is a global phenomenon that is significantly transforming public health patterns, including the emergence and spread of fungal infections such as IC. The sustained increase in atmospheric and ocean temperatures, together with changes in humidity levels and the growing frequency of extreme weather events, is altering the ecological behavior of fungi, favoring the adaptation and emergence of thermotolerant and resistant pathogenic strains [115,116]. From an environmental perspective, global warming has altered the distribution and survival of fungi in habitats that were previously hostile to their development. This transformation not only affects fungal diversity and the nutrient cycle, but more worryingly, it has created conditions favorable for the natural selection of strains capable of surviving increasingly high temperatures [117].

One of the most relevant hypotheses to explain the emergence of *C. auris* suggests climate change as a key evolutionary driver. In 2019, Casadevall and colleagues proposed the “thermal selection hypothesis,” according to which progressive environmental warming has favored the adaptation of certain environmental fungi to higher temperatures, allowing them to overcome the thermal barrier that historically limited their ability to infect humans [108]. For Latin America, this remains speculative and requires region-specific ecological and clinical analyses.

This phenomenon is not exclusive to *C. auris*, as other emerging strains, such as *Cryptococcus deuterogattii*, have also shown similar patterns of adaptation [118,119].

In addition to climate change, other environmental and anthropogenic factors have been hypothesized to be associated with the emergence of *Candidozyma auris* and other resistant fungi. The indiscriminate use of antifungals in agriculture, especially azoles, has exerted selective pressure on environmental microorganisms. Sharma and Kadosh (2023) found that these compounds, widely used in fruit and vegetable cultivation, may have contributed to the emergence of resistant strains in the environment, which subsequently migrate to the hospital environment [120]. This hypothesis of thermal selection has been proposed in other emerging fungi [118,121].

Another emerging hypothesis suggests the possible role of wild fauna as a passive vector in the spread of resistant fungi. From a One Health perspective, García-Bustos and colleagues proposed that migratory birds could act as carriers of *C. auris*, facilitating its transit from natural environments to rural or urban areas, where the fungus could establish itself and eventually enter the hospital environment. This integrative approach, which links human, animal, and environmental health, is essential for understanding the complex web underlying emerging mycoses [23,77,122].

Forced migration, overcrowding, shortages of drinking water, and limited access to healthcare in refugee camps significantly increase the vulnerability of these populations to invasive fungal infections. Added to this is the threat to food security, exacerbated by climate change, which can lead to malnutrition and, with it, a marked increase in the incidence of infectious diseases, including fungal infections. In addition, extreme weather events such as floods create humid environments conducive to the germination and spread of fungal spores. These conditions favor accelerated fungal growth, facilitating the large-scale dispersion of spores and mycotoxins [112,117,118,119,123,124].

Overall, it is recognized that various factors contribute to the emergence of invasive mycoses such as *C. auris*. These include the indiscriminate use of antifungals in agriculture, water contamination, inadequate hospital practices, and forced population displacement. Added to this are the interactions between degraded ecosystems, wild fauna, and the growth of vulnerable human populations, creating a scenario favorable to the emergence and spread of resistant pathogenic fungi [107,108,111,112,120,122].

## 7. Access to Health Services and Inequalities in the Region

Socioeconomic inequalities and differences in access to healthcare services in Latin America have a direct and significant impact on the management of fungal infections, including IC. The gap between public and private healthcare systems, as well as between urban and rural areas, and between countries with different levels of development, leads to profoundly unequal clinical scenarios. In many cases, the patient’s prognosis depends more on their geographical location or socioeconomic status than on the natural course of the disease [35,36,37,38,61]. In Brazil, for example, studies have shown that the incidence of candidemia in public hospitals is more than double that in private institutions, with rates of 2.42 versus 0.91 cases per 1000 ICU admissions, respectively. In addition, the distribution of strains differs depending on the type of institution: while *C. parapsilosis* predominates in public hospitals, *N. glabrata* is more common in private hospitals [34,61,97].

Access to antifungal treatments in Latin America also presents marked inequalities. In many countries in the region, FCZ and AmB deoxycholate are the only widely available antifungals, even though they are not always the most appropriate therapeutic option. First-line drugs, such as echinocandins or lipid formulations of AmB, have limited availability, particularly in public hospitals and rural areas. Meanwhile, 5-flucytosine—essential for the treatment of certain serious fungal infections—is available in only 20% of the hospitals evaluated [54]. The prices of antifungal drugs also vary considerably between countries in the region, with costs ranging from USD 1 to USD 31 per unit of FCZ. This variation can be a significant obstacle for many patients, especially when healthcare systems do not provide comprehensive coverage for antifungal treatment [125]. In Colombia, for example, the healthcare system largely covers the costs of antifungal treatments. However, the availability of medications remains uneven. While cities such as Bogotá and Medellín have a more adequate supply, many rural regions face critical shortages, limiting timely access to effective therapies [126].

In the context of IC, a key factor is the continuing education of healthcare personnel. In centers with greater resources, access to specialized training in fungal infections is more common, which translates into more appropriate clinical management by professionals. In contrast, in settings with less access to training, significant knowledge gaps persist. In this scenario, virtual education strategies (e-learning) become highly relevant for expanding access to training. However, there are barriers that can limit its effective implementation, such as poor internet connectivity, limited proficiency in other languages, and lack of knowledge or low familiarity with technological tools by some healthcare professionals [127,128,129,130].

In response to this situation, various regional initiatives have been launched to reduce existing gaps. The Pan American Health Organization (PAHO) and the Global Action Fund for Fungal Infections (GAFFI) have established strategic alliances to improve access to essential antifungals, standardizing clinical protocols, and strengthening the diagnostic capabilities of health systems in the region [131]. Telemedicine and virtual education are also promoted as key tools for training healthcare personnel in remote areas. However, these efforts face various challenges, including poor connectivity, language barriers, and unequal access to digital technologies, which limit their effective implementation in some regions [130,132].

## 8. Impact of Housing and Environment on IC in Latin America

Although IC has been extensively studied in hospital settings, its relationship with environmental factors and living conditions in Latin America remains poorly explored. Unlike other invasive mycoses, such as histoplasmosis or cryptococcosis, in which the link with the natural environment is well documented, candidiasis has traditionally been considered a nosocomial infection. However, recent data suggest that the physical environment and living conditions may also influence the vulnerability of certain population groups to this infection [133,134]. Although *Candida* is not a typical environmental fungus like *Aspergillus* or *Cryptococcus*, its presence has been documented in ecological niches outside the human environment. For example, strains such as *C. albicans* have been isolated in soil and vegetables, *C. tropicalis* in rivers and agricultural areas, and *P. kudriavzevii* in animals. In the particular case of *C. auris*, its detection in freshwater and marine bodies in coastal areas of Colombia reinforces the hypothesis that certain environmental settings could act as reservoirs or points of contact with humans, especially in populations with limited sanitary conditions [135,136].

One of the most important aspects from a social and environmental point of view is the quality of housing. In many urban areas of Latin America, informal settlements and marginalized neighborhoods lack basic services such as drinking water, adequate ventilation, and sanitation. These conditions favor the proliferation of opportunistic microorganisms and can facilitate colonization by yeasts, especially in humid or contaminated environments. Although the direct relationship between these conditions and IC has not yet been fully demonstrated, overcrowding and poor housing conditions have been associated with an increased risk of superficial infections, such as oral and vulvovaginal candidiasis. In addition, the high prevalence of chronic noncommunicable diseases in contexts of poverty—such as diabetes mellitus and chronic kidney diseases—significantly increases the risk of invasive *Candida* infections [135,136].

### Socio-Environmental Hypotheses and Research Gaps

The influence of the environment on invasive aspergillosis is closely related to exposure to spores present in the air, especially in contexts where construction work is being carried out or where ventilation systems are inadequate. These factors have been implicated in hospital outbreaks as they facilitate the spread of *Aspergillus* conidia in clinical areas [16,38]. In addition, there are isolated studies on other mycoses that support this perspective. In Brazil, for example, *Cryptococcus gattii* has been isolated from household dust in wooden houses located in the Amazon region, demonstrating that indoor spaces can also act as reservoirs for pathogenic fungi. Although such a direct link has not been established in the case of *Candida*, moisture accumulation and environmental contamination could play a role in its spread or in increasing the host’s vulnerability [137].

Although global warming has been proposed as a selective pressure that enables *Candida auris* to breach the mammalian thermal barrier, robust climate–disease correlations for Latin America remain scarce. Nonetheless, the recent isolation of *C. auris* from Colombian brackish estuaries indicates that comparable ecological niches are present in the region [109]. We therefore regard both climatic and socio-structural pressures as plausible yet unconfirmed contributors that deserve focused investigation rather than definitive explanatory variables.

## 9. Conclusions

IC has become a significant threat to public health in Latin America. Its increasing incidence, especially in ICUs, and the high mortality rate associated with it highlight a clinical urgency that cannot be ignored. Although IC has historically been attributed to *C. albicans*, there is currently an epidemiological shift toward non-*albicans* strains, many of which are resistant to the most commonly used antifungal agents.

Latin America faces a singular convergence of factors driving invasive fungal disease: the rapid expansion of Clade IV *Candidozyma auris*, the escalation of azole resistance in *Candida parapsilosis*, persistent diagnostic shortfalls in resource-limited hospitals, and the destabilizing effects of migration and fragmented health services.

The emergence of *C. auris*, characterized by its remarkable ability to spread, resistance to multiple antifungals, and persistence in the environment, has highlighted the structural weaknesses of health systems in the region. Gaps in diagnosis, the limited availability of effective therapies, and unequal access to medical care have left many populations in a vulnerable situation, especially in public health systems, rural areas, or impoverished communities.

In addition to these clinical and technical issues, social and environmental factors may amplify the impact of fungal infections.

Climate change, the widespread use of antifungals in agriculture, inadequate housing conditions, and structural inequalities in healthcare systems may contribute to creating an environment favorable to the emergence of resistant pathogens and the spread of serious fungal infections; however, direct evidence for these associations in Latin America is scarce, and their relevance in this region remains a plausible but unconfirmed hypothesis, informed partly by observations from other geographic settings.

To address this challenge, it is essential to adopt a comprehensive and coordinated approach that transcends the strict hospital setting. It is necessary to strengthen mycological surveillance systems, ensure equitable access to advanced diagnostic tests and state-of-the-art antifungals, and train health personnel at all levels of care.

Regional research must be intensified, especially in countries where information on IC is limited, to obtain a more accurate picture of its actual burden and local characteristics. Only on this basis will it be possible to design effective, contextualized, and evidence-based public policies. It is time to look beyond the microscope and move toward solutions that integrate science, public health, and social justice.

## Figures and Tables

**Figure 1 jof-11-00609-f001:**
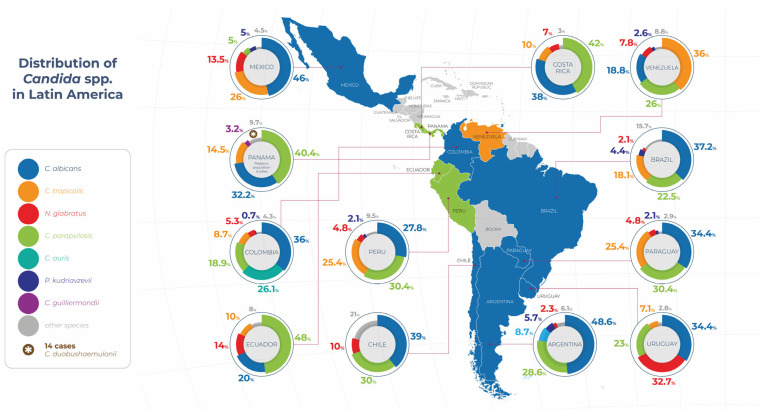
Distribution of *Candida* spp. in Latin America. Data are drawn from published studies originating from each country [15,20,21,22,23,24,25,26,27,28,29,30,31,32].

**Figure 2 jof-11-00609-f002:**
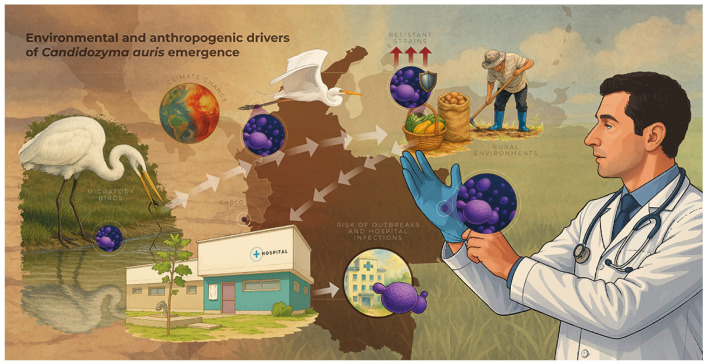
Environmental and anthropogenic drivers of *Candida auris* emergence.

**Table 1 jof-11-00609-t001:** Comparison of diagnostic tests for IC in Latin America.

Diagnostic Method	Main Advantages	Main Limitations	*C. auris*	Availability in Latin America	References
Blood culture (biochemical tests (VITEK, API, and microscanning))	- Reference standard: isolation, identification, and susceptibility testing	- Prolonged duration (3–4 days) - Moderate sensitivity (21–71%)	Grows reliably at 37–40 °C but identification by conventional phenotypic methods often misidentifies *C. auris*; requires species-level confirmation	Widely available in most hospitals	[57,58]
1,3-β-D-glucan (serum)	- High sensitivity for invasive candidiasis (92%) - Useful for ruling out systemic infection	- Low specificity (positive in other mycoses) - High cost	Sensitivity ~71% for *C. auris* candidemia (lower than other *Candida* spp.); frequent false negatives; not recommended as a standalone diagnostic marker for *C. auris*	Absent in most centers	[57,58,59]
Mannan/Antimannan (serum)	- Allows detection of circulating *Candida* antigens	- Reduced sensitivity (55–60%) - Interlaboratory variability	Very low sensitivity (<60%) and high inter-laboratory variability for non-albicans species; *C. auris* often yields negative or indeterminate results; rarely used for *C. auris* detection	Very scarce: infrequent clinical use	[57,58,60]
Quantitative PCR (in blood)	- Results can be obtained in a few hours—sensitivity 92%; specificity 95%	- Requires reference laboratory and highly trained personnel	Direct detection of *C. auris* DNA in blood within 3–6 h; sensitivity ~92–93%, and specificity ~95–96%; requires specialized equipment and trained personnel; available in ~20% of reference laboratories in Latin America	Only 20% of centers in Argentina	[57,58,61]
Conventional endpoint PCR + Sanger	Confirmation of amplicon size via gel electrophoresis, and enhanced sensitivity and specificity through Sanger sequencing of gel-purified products—especially beneficial for detecting longer DNA fragments that short-amplicon qPCR assays may overlook	- Longer turnaround time than qPCR - Requires sequencing capacity	Gold-standard molecular confirmation of *C. auris* via ITS or D1/D2 sequencing; 100% specificity; turnaround of 1–2 days; limited to labs with sequencing capacity; ideal for resolving ambiguous identifications	Variable; typically available only in reference or specialized labs	[57,58,62]
Metagenomic next-generation sequencing (mNGS)	- Broad, hypothesis-free pathogen detection - High sensitivity and specificity for fungi in multiple studies	- High cost, need for specialized infrastructure, bioinformatics expertise, potential contamination, longer turnaround	Culture-independent detection of *C. auris* (and co-pathogens) with high sensitivity and specificity directly from clinical samples; turnaround of 1–2 days	Limited; available in few references’ centers	[57,58,63]
T2*Candida* (magnetic resonance imaging)	- Direct and rapid detection (3–5 h) - Sensitivity 91%; specificity 99	- Very high cost - Only five specific strains	Current commercial panel does not include *C. auris* and thus fails to detect this species; an investigational T2 *C. auris* panel exists but is not yet clinically available	Almost unavailable; very limited data	[57,58,64]
Matrix-assisted laser desorption/ionization time-of-flight mass spectrometry	- MALDI-TOF is highly accurate (97–99%) for common species	- Expensive equipment - Requires previous cultivation	When using an updated library, it accurately distinguishes *C. auris* from related yeasts (>90% correct ID) within minutes of colony growth; requires prior culture	Available in 20–50% of high-volume laboratories [59]	[57,58,65]

## Data Availability

No new data were created or analyzed in this study. Data sharing is not applicable to this article.

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
