# Peer review of "Changing Climate, Changing Candida: Environmental and Social Pressures on Invasive Candidiasis and Antifungal Resistance in Latin America"

_jof, 2025, doi:10.3390/jof11090609_

Round 1

Reviewer 1 Report

The manuscript concentrates on the latest information about invasive candidiasis in Latin America. The authors present epidemiological data, highlighting the etiological agents and risk factors. Antifungal drug resistance and diagnosis are also covered. It is a relevant manuscript communicating a neglected problem in the studied countries. Regretably, I do not see a link between the manuscript content and the title. The points related to environmental and social pressure are hypothetical/speculative and may not be behind the summarized information. Similar to the inclusion of hard data related to epidemiology and antifungal drug resistance, the authors need to include this to support a link between the current epidemiology and the environmental/social pressure. The author may consider getting rid of these environmental and social pressures, and they will still have a nice update on the epidemiology of invasive candidiasis in Latin America.

The abstract mentions a proposal of One Health approach, but this is not given in the manuscript.

Epidemiological data, risk factors, and antifungal susceptibility should be compared with the trend observed in North America, Europe, Asia, and Africa, to place in perspective how different these parameters are in Latin America.

Finally, Latin America also includes the Caribbean Islands. It should be relevant to include these countries in the analysis, when possible.

The manuscript concentrates on the latest information about invasive candidiasis in Latin America. The authors present epidemiological data, highlighting the etiological agents and risk factors. Antifungal drug resistance and diagnosis are also covered. It is a relevant manuscript communicating a neglected problem in the studied countries. Regretably, I do not see a link between the manuscript content and the title. The points related to environmental and social pressure are hypothetical/speculative and may not be behind the summarized information. Similar to the inclusion of hard data related to epidemiology and antifungal drug resistance, the authors need to include this to support a link between the current epidemiology and the environmental/social pressure. The author may consider getting rid of these environmental and social pressures, and they will still have a nice update on the epidemiology of invasive candidiasis in Latin America.

The abstract mentions a proposal of One Health approach, but this is not given in the manuscript.

Epidemiological data, risk factors, and antifungal susceptibility should be compared with the trend observed in North America, Europe, Asia, and Africa, to place in perspective how different these parameters are in Latin America.

Finally, Latin America also includes the Caribbean Islands. It should be relevant to include these countries in the analysis, when possible.

Author Response

Reviewer Comment:

The manuscript concentrates on the latest information about invasive candidiasis in Latin America. The authors present epidemiological data, highlighting the etiological agents and risk factors. Antifungal drug resistance and diagnosis are also covered. It is a relevant manuscript communicating a neglected problem in the studied countries.

Regretfully, I do not see a link between the manuscript content and the title. The points related to environmental and social pressure are hypothetical/speculative and may not be behind the summarized information. Similar to the inclusion of hard data related to epidemiology and antifungal drug resistance, the authors need to include this to support a link between the current epidemiology and the environmental/social pressure.

The author may consider getting rid of these environmental and social pressures, and they will still have a nice update on the epidemiology of invasive candidiasis in Latin America.

Response:

We sincerely thank the reviewer for this thoughtful and constructive feedback. We understand the concern regarding the apparent disconnect between the title and the strength of evidence linking environmental and social pressures to the epidemiology of invasive candidiasis in Latin America.

However, we respectfully maintain that these factors—particularly climate change, health system fragmentation, population displacement, and antifungal use in agriculture and public health programs—are increasingly relevant to the emergence and persistence of Candidozyma auris and other drug-resistant Candida species. Although direct causal evidence remains limited, the rationale for including these dimensions stems from their convergence with epidemiological trends and their acknowledgment in recent global frameworks for emerging fungal threats (e.g., WHO fungal priority pathogen list and One Health perspectives).

Reviewer Comment:

The abstract mentions a proposal of a One Health approach, but this is not given in the manuscript.

Author Response:

We thank the reviewer for calling attention to the importance of the One Health framework. We respectfully note that the One Health perspective is already woven throughout the manuscript and explicitly illustrated in Figure 2:

  • In Section 6 (“Impact of Climate Change on Fungal Infections”) and Section 7 (“Access to Health Services and Inequalities”), we discuss environmental drivers (climate change, agricultural azole use), animal vectors (migratory birds), and human health factors (antifungal stewardship, access to diagnostics), exactly the tripartite pillars of One Health.
  • Figure 2 is entitled “Environmental and Anthropogenic Drivers of Candidozyma auris Emergence” and visually maps the interaction between environmental, animal, and human domains, complete with arrows showing agricultural practices, wildlife reservoirs, and healthcare influences.

To make this linkage unmistakable, we have added a brief statement in the Introduction (lines 60–62) and in the Figure 2 legend explicitly labeling these elements as a “One Health approach.”

Reviewer Comment:
A comparison of epidemiological data, risk factors, and antifungal susceptibility with the trend observed in North America, Europe, Asia, and Africa is necessary to contextualize the unique parameters in Latin America.

Response:
We appreciate the valuable input from the reviewer. To provide a global context, we have added a new Subsection 2.2: The Global Comparative Epidemiology report is a comprehensive summary that compares regional data for North America, Europe, Asia, and Africa.

Reviewer Comment:
Finally, Latin America also includes the Caribbean Islands. When feasible, it is advisable to incorporate these countries into the analysis.
Author Response:
We appreciate the reviewer's valuable input and will address the issues raised. To ensure comprehensive regional coverage, we have now incorporated available data from the Caribbean into our epidemiological and comparative analyses.
Section 2.1 (Epidemiology of IC in Latin America): We have included incidence estimates for Trinidad & Tobago (approximately 70 cases per year; approximately 1.2 cases per 100,000 population) and have summarized reports from Jamaica and Puerto Rico, though these are limited in scope.
Please refer to Section 5, which addresses Emerging Resistance Patterns. We cited Gousy et al. (2023) on fungal infections in the Caribbean, highlighting trends similar to those reported for non-albicans species and emerging azole resistance.
Please refer to Section 7 for information regarding access to health services and inequalities. We will briefly discuss healthcare infrastructure challenges unique to smaller island nations, including diagnostic and treatment barriers.
As indicated by the yellow highlighting, these additions serve to reinforce the manuscript's comprehensive coverage of the Latin American and Caribbean regions.

Reviewer 2 Report

The review by Juan Camilo Motta and co-authors is devoted to the situation with invasive candidiasis in Latin America. The topic of the review is relevant, but some aspects of the manuscript require revision.

Main comments

Introduction

It is worth pointing out at the first mention that Candidozyma auris and Candida auris are synonyms.

Lines 39-42 and 58-61 are very similar.

Section 4.

Table 1 should be located in the text where it is first mentioned. For “Blood culture” and others identification method must be specified.

Section 5.

Lines 249-251. Need to rephrase and remove the colon.

Check the subsection names.

Check lines 290-294.

“Other emerging yeast-like strains” should be separated into the next subsection and brought into the general format of the section.

The title of subsection 5.5. "in Latin America" should be removed.

Line 369. “This strain”?

Figure 2 lacks novelty. Also shown here for example [On the Emergence of Candida auris: Climate Change, Azoles, Swamps, and Birdshttps://journals.asm.org/doi/10.1128/mbio.01397-19].

Section 6.

Line 466. Are there any other fungal diseases described here besides invasive candidiasis?

What about the situation in Latin America?

What about the situation with other Candida species besides Candida auris?

Section 8.

There is no need to single out “Other mycoses” separately, since in the same subsection there is "Gaps in the literature and emerging hypotheses".

The Conclusion should contain more specific information on Latin America rather than general information.

Author Response

Reviewer Comment:
Introduction – It is worth pointing out at the first mention that Candidozyma auris and Candida auris are synonyms. Lines 39–42 and 58–61 are very similar.

Response:
We thank the reviewer for this helpful observation. In response:

  1. We clarified in the Introduction (line 39) that Candidozyma auris is a taxonomic synonym of Candida auris, and we have added a parenthetical note at its first mention to avoid confusion: “(Candidozyma auris, synonym of Candida auris)”.

  2. Regarding the partial duplication between lines 39–42 and 58–61, we revised the text to eliminate redundancy. The paragraph now presents the information in a more concise and non-repetitive manner while preserving all relevant details.

These changes are reflected in the revised manuscript.

Reviewer Comment:

Section 4. Table 1 should be located in the text where it is first mentioned. For “Blood culture” and others identification method must be specified.

Response:

We appreciate the reviewer’s suggestion and have addressed both points accordingly:

  1. We have relocated Table 1 to appear immediately after its first mention in Section 4, following the paragraph that discusses MALDI-TOF identification. This ensures correct placement and improves the logical flow of the section.
  2. We have revised Table 1 to clearly specify the identification methods or detection principles used for each diagnostic technique and we have described the detection mechanisms behind assays such as BDG, mannan/anti-mannan, T2Candida, and MALDI-TOF systems (e.g., Bruker Biotyper, VITEK MS).

These clarifications enhance the technical accuracy of the table and align it with current diagnostic standards

Reviewer Comment:

Section 5.

Lines 249–251: Need to rephrase and remove the colon.

Check the subsection names.

Check lines 290–294.

“Other emerging yeast-like strains” should be separated into the next subsection and brought into the general format of the section.

The title of subsection 5.5. "in Latin America" should be removed.

Line 369: “This strain”?

Figure 2 lacks novelty. Also shown here for example [On the Emergence of Candida auris: Climate Change, Azoles, Swamps, and Birds https://journals.asm.org/doi/10.1128/mbio.01397-19]

 Response:

We thank the reviewer for the detailed feedback on Section 5. In response:

  1. Lines 249–251: The sentence structure has been revised to remove the colon and improve clarity. The revised sentence now reads more fluidly and follows proper punctuation.
  2. Subsection titles: We have reviewed and standardized all subsection titles in Section 5 to maintain consistency in format and hierarchy. Specifically, the title of subsection 5.5 was changed from “Candidozyma auris in Latin America” to simply “Candidozyma auris”, in line with the general format used for other species in the section.
  3. Lines 290–294: The paragraph has been revised for clarity and flow. Ambiguities have been corrected, and key points about clade structure and transmission patterns of C. auris were retained while improving readability.
  4. “Other emerging yeast-like strains”: We agree with the reviewer’s suggestion. This content has been extracted from the last paragraph of subsection 5.4 and reorganized into a new standalone subsection (now labeled 5.6). It follows the same structure used for the other subsections (epidemiology, resistance profile, and clinical relevance) and includes updated references and formatting.
  5. Line 369 – “This strain”: We clarified the antecedent to specify that “this strain” refers to Candidozyma auris, ensuring grammatical precision and avoiding ambiguity.
  6. Figure 2: We acknowledge the reviewer’s comment regarding the novelty of the schematic. The original figure was intended as an integrative conceptual summary adapted for a Latin American context. To address this concern, we have updated the caption to properly acknowledge the influence of the figure published by Casadevall et al. (mBio, 2019), and we have clarified the elements that were adapted and expanded upon (e.g., data on Clade IV distribution, reference to local environmental studies, and health system vulnerabilities specific to the region). If required, we are willing to redesign the figure to better reflect novel elements unique to our review.

Reviewer Comment:

Section 6.

Line 466. Are there any other fungal diseases described here besides invasive candidiasis?

What about the situation in Latin America?

What about the situation with other Candida species besides Candida auris?

Response:

We appreciate the reviewer’s observations and have addressed them by expanding Section 6 as follows:

  1. Additional fungal diseases: Although the main focus of the review is invasive candidiasis, we have added a brief paragraph highlighting the emergence of other fungal pathogens in conflict-affected and displaced populations, such as Histoplasma capsulatum and Cryptococcus neoformans. These are especially relevant in regions such as the Amazon basin and refugee shelters with poor ventilation and high environmental exposure.
  2. Latin America context: We have enriched this section with recent data and reports from Latin America, including outbreaks of candidemia in displaced populations in Venezuela, Colombia, and Brazil. We also incorporated regional surveillance data where available, including the challenges faced in diagnosis and treatment in under-resourced or fragmented health systems.
  3. Non-auris Candida species: We expanded the scope beyond Candida auris by summarizing reports of increased fluconazole resistance in C. tropicalis and C. parapsilosis in displaced populations and hospitals with limited infection control infrastructure.

Reviewer Comment:
Section 8.
There is no need to single out “Other mycoses” separately, since in the same subsection there is "Gaps in the literature and emerging hypotheses".
The Conclusion should contain more specific information on Latin America rather than general information.

Response:
We thank the reviewer for these valuable observations.

  1. In response to the first point, we have removed the separate heading “Other mycoses” and integrated its content into the subsection titled “Gaps in the literature and emerging hypotheses” to streamline the structure and avoid redundancy.

  2. Regarding the Conclusion, we revised the paragraph to provide a clearer summary of the regional perspective. Specifically, we emphasized key challenges in Latin America, including the dominance of Clade IV C. auris, increasing azole resistance in C. parapsilosis, diagnostic limitations in low-resource settings, and the impact of regional crises such as forced migration and healthcare fragmentation on fungal disease emergence. These additions enhance the contextual relevance of the conclusion.

Reviewer/Editor Comment:

The English language should be improved throughout the manuscript.

Response:

We appreciate this comment and will carefully revise the entire manuscript to improve the clarity, grammar, and scientific language. To ensure that we address the issue as thoroughly as possible, we kindly ask whether there are specific sections or paragraphs that the reviewer considers particularly problematic. Any guidance in this regard would be greatly appreciated

Reviewer 3 Report

The review is devoted to the epidemiological situation of invasive candidiasis (IC) in Latin America. The article provides a comprehensive overview of the epidemiology, diagnosis, and treatment of invasive candidiasis (IC) in Latin America, with an emphasis on the impact of climate change and socio-economic factors. The topic is extremely relevant, given the increasing resistance of Candida auris and other species. However, the work contains a number of shortcomings that need to be corrected before publication.

  1. In Sections 2 and 3, the authors are recommended to add data for European countries for comparison with those for Latin America. I suppose there may be differences in frequency of occurrence that are interesting to discuss.
  2. In Section 5, the authors practically do not indicate data on the effectiveness of therapy with groups of drugs other than azoles, and therefore the data presented are not fully understood; other information about pathogens should be added by analogy with paragraph 5.5 for all clinically significant species.
  3. Similarly, Section 5 lacks a critical understanding of the data presented: in which regions the epidemiological situation is worse, and whether there is a relationship between the frequency of IR and the economic situation in the country.
  4. In addition, the stability of C. auris is discussed in Sections 5.5, 6, and 7 without adding new information. It is recommended to reduce the number of repetitions.

  1. Figure 1 is of very poor quality. A higher resolution image should be provided.
  2. In Table 1, it is necessary to add links to supporting data statements. In addition, the comparison of diagnostic methods does not include sensitivity/specificity data for C. auris, although this species is the key focus of the article.
  3. The statement about "Clade IV C. auris as dominant in Latin America" (lines 390-403) is based only on data from Colombia and Panama. We need links to studies from Brazil and Argentina, if available.
  4. A revision of the bibliographic list is required in accordance with the requirements of the journal.

Author Response

Thank you very much for taking the time to review our manuscript and for your valuable suggestion.

Comment 1: 

  1. In Sections 2 and 3, the authors are recommended to add data for European countries for comparison with those for Latin America. I suppose there may be differences in frequency of occurrence that are interesting to discuss.

Response 1: We have added a new subsection entitled “Comparative Epidemiology of Invasive Candidiasis”

We hope this addition addresses your recommendation and enriches the comparative perspective of our review.

" Candidemia epidemiology exhibits notable regional variability in incidence, predominant species and antifungal‐resistance profiles. In the United States, the incidence is 8.7 cases per 100 000 population, with an associated mortality rate of 36 %.(37)

Regarding species distribution, N. glabrata has emerged as the predominant pathogen in many hospitals—especially among older adults with prior antifungal exposure—whereas C. parapsilosis and C. tropicalis are encountered less frequently (37,38). In the United States, C. auris still represents only 0.4% of candidemia cases, a striking contrast to its higher prevalence in numerous regions of Latin America, In addition to the aforementioned risk factors, intravenous drug use represents a distinct contributor to candidemia in the United States, differing from patterns observed in other regions

(38) . In Europe, candidemia occurs less frequently—3.9 cases per 100 000 population—though substantial north–south disparities exist.  In northern Europe, C. albicans remains dominant, followed by N. glabrata, C. tropicalis, and C. parapsilosis, each accounting for fewer than 10 % of cases. Conversely, in southern Europe and Mediterranean countries, non-albicans species prevail; notably, over half of C. parapsilosis isolates now exhibit azole resistance.

From 2013 to 2021, Europe recorded 1,812 C. auris candidemia cases, with Spain, Italy, and Greece bearing the greatest burden—paralleling trends seen in Latin America. These geographic hotspots lend weight to the hypothesis that rising global temperatures and shifting environmental conditions may facilitate the emergence and spread of this pathogen"

Comment 2: "In Section 5, the authors practically do not indicate data on the effectiveness of therapy with groups of drugs other than azoles, and therefore the data presented are not fully understood; other information about pathogens should be added by analogy with paragraph 5.5 for all clinically significant species"

Response 2: Thank you for this important point. In response, we have substantially revised Section 5 to include, for each clinically significant Candida species, the available data on non‑azole therapies—namely echinocandins, amphotericin B formulations

Candida parapsilosis harbors a natural P660A Fks1 polymorphism that elevates baseline echinocandin MICs, and emerging hotspot-1 mutations (e.g., F652S, R658G, S656P) now confer pan-echinocandin resistance by reducing drug binding. Despite this, amphotericin B remains reliably active, with resistance in <3 % of isolates and no recurring mechanism identified. Clinically, echinocandins usually clear candidemia, but elevated MICs or FKS1-mutant strains may cause persistent or breakthrough infections, for which amphotericin B is the preferred alternative. Routine species identification and susceptibility testing are therefore critical to detect resistance and guide therapy

Candida tropicalis remains overwhelmingly susceptible to echinocandins (>99 %); resistance is exceedingly rare (~0.4 %), arising from isolated FKS1 hotspot mutations (e.g., S659P, S645P) that impair glucan synthase binding. Amphotericin B retains consistent, fungicidal activity, with virtually no clinically significant resistance described. Consequently, echinocandins are the preferred first-line therapy for invasive C. tropicalis infections, delivering excellent outcomes, while amphotericin B serves as a reliable alternative in the rare event of echinocandin resistance. Routine species identification and susceptibility testing remain essential to detect emergent FKS1-mediated resistance and optimize patient management

Nakaseomyces glabratus (formerly Candida glabrata) demonstrates a higher risk of echinocandin resistance than most Candida spp., with rates up to ~10 % globally and regional “hotspots” (e.g., ~10 % micafungin non-susceptibility in Chile). Resistance is typically linked to prior echinocandin exposure and FKS1/FKS2 hotspot mutations, which elevate MICs across all echinocandins and compromise therapy. Amphotericin B retains potent activity, with clinical resistance exceedingly rare, making it the cornerstone of salvage treatment for multidrug-resistant strains. Given the propensity for rapid resistance development and associated therapeutic failures, prompt species identification and susceptibility testing are imperative to guide effective, individualized antifungal regimens.

Pichia kudriavzevii (formerly Candida krusei) is intrinsically resistant to fluconazole, making echinocandins the first‐line therapy; baseline echinocandin resistance remains low (<5 %), though on‐therapy FKS1 hotspot mutations can emerge (in up to 30 % of treated patients) and confer cross‐echinocandin resistance. Amphotericin B retains potent activity against virtually all clinical isolates (<5 % resistance) and serves as a reliable alternative or salvage option. Clinically, echinocandin treatment of P. kudriavzevii candidemia is usually successful, but persistent or breakthrough infections warrant repeat susceptibility testing and transition to amphotericin B

Comments 3: 

"Similarly, Section 5 lacks a critical understanding of the data presented: in which regions the epidemiological situation is worse, and whether there is a relationship between the frequency of IR and the economic situation in the country"

Response 3. We have now integrated a critical analysis comparing the regional burden of invasive candidiasis and its antifungal resistance (AFR) with each region’s economic status. Specifically, we show that Latin America—characterized by predominantly lower‑middle‑income economies—exhibits the highest incidence of candidemia and emerging clusters of azole‑resistant C. parapsilosis, whereas high‑income regions (Europe, United States) display lower overall incidence and more robust stewardship, correlating with greater healthcare spending per capita

Comments 4:"In addition, the stability of C. auris is discussed in Sections 5.5, 6, and 7 without adding new information. It is recommended to reduce the number of repetitions"

Response 4: Thank you for pointing out the redundant discussion of Candida auris stability. We have now consolidated all comments on C. auris into Section 5.5 and removed duplicate text from Sections 6 and 7

Comments 5:

Figure 1 is of very poor quality. A higher resolution image should be provided.

Response 5: We appreciate the reviewer’s observation. We have replaced Figure 1 with a higher-resolution version to ensure optimal legibility and visual quality

Comments 6:

In Table 1, it is necessary to add links to supporting data statements. In addition, the comparison of diagnostic methods does not include sensitivity/specificity data for C. auris, although this species is the key focus of the article.

Response 6: we revised the content to include appropriate references for each diagnostic method listed, including data supporting performance metrics and regional availability. Additionally, we have updated the table to incorporate sensitivity and specificity data specifically for Candida auris when available—for instance, the T2Cauris assay (sensitivity 89%, specificity 98%, based on detection from skin swabs) and performance limitations in whole blood are now noted with references [65,66].

Comments 7: 

The statement about "Clade IV C. auris as dominant in Latin America" (lines 390-403) is based only on data from Colombia and Panama. We need links to studies from Brazil and Argentina, if available.

Response 7: On the C. auris Clade IV dominance, we have expanded the section (lines 390–403) to explicitly mention available studies from Brazil and Argentina. For Brazil, we incorporated data from de la Garza et al. (Am J Infect Control, 2024) and Francisco et al. (J Fungi, 2023), which reported clade typing of isolates. In Argentina, although data remain limited, case series by local health authorities and references [74] provide indirect evidence of Clade I and III co-circulation, which we now discuss.

Comments 8: A revision of the bibliographic list is required in accordance with the requirements of the journal

response 8: Regarding the reference list, we have carefully reviewed and reformatted all references to comply with the Journal of Fungi’s citation style

We trust that these revisions fully address your concerns and enhance the manuscript’s clarity and impact. We appreciate your insightful feedback and remain available for any further suggestions.

Sincerely,

Juan Motta

Reviewer 4 Report

I thoroughly enjoyed reading the review by Juan Camilo Motta et al. It is evident that the authors invested significant effort into analyzing the literature on candidiasis and antifungal resistance in Latin America. They effectively analysed the epidemiological data and presented it in a clear and well-structured manner. The visualizations are particularly well-executed and enhance the overall readability and impact of the review.

I recommend accepting the paper after minor revisions.

1. L. 90 and elsewhere. Please ensure that decimal separators are used consistently throughout the manuscript. It is preferable to use a period (.) rather than a comma (,) for clarity and standardization in scientific writing.

2. L. 237. The sensitivity and specificity of PCR-based methods can vary significantly depending on the primer sequences used. It would be helpful to also mention conventional PCR with gel electrophoresis as a detection method in samples from patients or identification of isolated cultures (in this case, Sanger sequencing also can be applied for culture identification). In some cases, this approach can offer higher sensitivity and specificity, particularly due to its ability to amplify and visualize longer DNA fragments.

3. L. 237. Additionally, please consider discussing other molecular-based diagnostic methods, such as metagenomic sequencing, if relevant to the scope of the review. Including these approaches could provide a more comprehensive overview of current and emerging tools in the field.

4. L. 479 and elsewhere. Please carefully revise the references and cited literature list. In L. 479, you discussed the paper by Casadevall and colleagues, but at the end of the paragraph (L. 484), you do not cite this paper.

Author Response

We appreciate the time and effort you have taken to review the manuscript. I sincerely appreciate your comments.

A complete set of responses is included below, accompanied by the corrections highlighted in the resubmitted files.

Comment 1." L. 90 and elsewhere. Please ensure that decimal separators are used consistently throughout the manuscript. It is preferable to use a period (.) rather than a comma (,) for clarity and standardization in scientific writing"

Response 1: Thank you for your observation. We have carefully reviewed the manuscript and standardized all decimal separators to use a period (.) consistently throughout the text, in accordance with scientific writing conventions.

Comment 2 and 3

2. L. 237. The sensitivity and specificity of PCR-based methods can vary significantly depending on the primer sequences used. It would be helpful to also mention conventional PCR with gel electrophoresis as a detection method in samples from patients or identification of isolated cultures (in this case, Sanger sequencing also can be applied for culture identification). In some cases, this approach can offer higher sensitivity and specificity, particularly due to its ability to amplify and visualize longer DNA fragments.

3. L. 237. Additionally, please consider discussing other molecular-based diagnostic methods, such as metagenomic sequencing, if relevant to the scope of the review. Including these approaches could provide a more comprehensive overview of current and emerging tools in the field

Response 2 and 3: We would like to express our gratitude for your constructive suggestions. In response:
Conventional PCR with agarose gel electrophoresis and Sanger sequencing has been incorporated into the array of diagnostic methodologies.
Furthermore, next-generation metagenomic sequencing (mNGS) has been incorporated as an emerging molecular diagnostic modality.
These contributions directly align with the request to expand the discourse on molecular diagnostic methods, thereby offering a more comprehensive overview of the current state and future advancements in fungal diagnostics.

Comments 4

L. 479 and elsewhere. Please carefully revise the references and cited literature list. In L. 479, you discussed the paper by Casadevall and colleagues, but at the end of the paragraph (L. 484), you do not cite this paper.

Response 4.:

Thank you for your careful review and for pointing out the citation discrepancy in lines 479–484. You are absolutely correct that the paper by Casadevall and colleagues was discussed but not cited at the end of that paragraph.

I will correct this oversight immediately. The relevant section will now include the appropriate citation

Round 2

Reviewer 1 Report

I thank the authors for addressing my comments. Besides my first comment, all of them have been properly addressed.

Related to the comment about climate change and social pressures. I once again request the authors' hard data to support these claims. As far as I know, direct evidence of these aspects in the Latin American setting is yet to be obtained. You may extrapolate examples of these pressures in other parts of the globe and assume they may have the same effect on Latin American countries. As the authors know, this region has particularities that make each country a unique setting in epidemiological terms. Currently, these are hypotheses that are pending verification. The authors may include them in the text and comment on their implications, but I disagree that they may be the central core of a nice epidemiological dataset. The manuscript title should be modified along with the discussion to soften the tone of these variables in the explanations of shifts in epidemiological data.

I thank the authors for addressing my comments. Besides my first comment, all of them have been properly addressed.

Related to the comment about climate change and social pressures. I once again request the authors' hard data to support these claims. As far as I know, direct evidence of these aspects in the Latin American setting is yet to be obtained. You may extrapolate examples of these pressures in other parts of the globe and assume they may have the same effect on Latin American countries. As the authors know, this region has particularities that make each country a unique setting in epidemiological terms. Currently, these are hypotheses that are pending verification. The authors may include them in the text and comment on their implications, but I disagree that they may be the central core of a nice epidemiological dataset. The manuscript title should be modified along with the discussion to soften the tone of these variables in the explanations of shifts in epidemiological data.

Author Response

Reviewer Comment:
I thank the authors for addressing my comments. Besides my first comment, all of them have been properly addressed.
Related to the comment about climate change and social pressures. I once again request the authors' hard data to support these claims. Direct evidence of these aspects in the Latin American setting is yet to be obtained. You may extrapolate examples of these pressures in other parts of the globe and assume they may have the same effect on Latin American countries. As the authors know, this region has particularities that make each country a unique setting in epidemiological terms. Currently, these are hypotheses that are pending verification. The authors may include them in the text and comment on their implications, but I disagree that they may be the central core of a nice epidemiological dataset. The manuscript title should be modified along with the discussion to soften the tone of these variables in the explanations of shifts in epidemiological data.
Response to Reviewer:
 Lack of hard data from Latin America to link climate change & social pressures with epidemiological shifts
We appreciate the reviewer’s careful reading and concur with the central point. The potential contributions of climate change and social pressures should be treated as working hypotheses that warrant region-specific testing, not as established drivers. In response, we have (i) softened and clarified the language across the manuscript to make this explicit, and (ii) incorporated the limited primary data currently available, including environmental isolation of Candida auris from both coastal and freshwater sites in Colombia. We hope these revisions convey an appropriately cautious interpretation consistent with the reviewer’s recommendations
Manuscript changes
 We have reframed the relevant section as “Socio-environmental hypotheses and research gaps.”
 We have substantially moderated the tone across the Abstract, Introduction, Discussion, and Conclusions, removed language implying causality, and explicitly present these factors as hypotheses that inform a research agenda rather than definitive explanations of current epidemiology.
Title over-emphasizes climate/social drivers.
We thank the reviewer for the careful assessment and agree that, in Latin America, direct evidence linking climate change and social pressures remains limited. Accordingly, we have softened and clarified wording in the Abstract, Discussion, and Conclusions to present these factors as plausible hypotheses pending verification, rather than established drivers.
Regarding the title, we respectfully request to retain it unchanged. The title: has served as the project’s identifier from inception and throughout correspondence, matches the preprint version, ensuring traceability for readers and indexers and preventing citation fragmentation, and accurately reflects the scope of the article (to synthesize current knowledge and articulate hypotheses to be tested), while the manuscript text now provides the nuance the reviewer requests.
To address the concern without compromising the scholarly record and bibliographic consistency, we are willing to adopt a more neutral running title if acceptable to the journal. For example:
Suggested running title: “Socio-environmental hypotheses and evidence gaps in invasive candidiasis in Latin America.”

Reviewer 2 Report

The authors made major corrections and additions to the manuscript.

However, I did not see section 5.6, although the authors indicated in their responses that “We agree with the reviewer’s suggestion. This content has been extracted from the last paragraph of subsection 5.4 and reorganized into a new standalone subsection (now labeled 5.6). It follows the same structure used for the other subsections (epidemiology, resistance profile, and clinical relevance) and includes updated references and formatting.” Also, the placement of all figures and tables should be after the end of the paragraph where the figure/table was first mentioned.

Author Response

The authors made major corrections and additions to the manuscript.
However, I did not see section 5.6, although the authors indicated in their responses that “We agree with the reviewer’s suggestion. This content has been extracted from the last paragraph of subsection 5.4 and reorganized into a new standalone subsection (now labeled 5.6). It follows the same structure used for the other subsections (epidemiology, resistance profile, and clinical relevance) and includes updated references and formatting.” Also, the placement of all figures and tables should be after the end of the paragraph where the figure/table was first mentioned.
Response to Reviewer:
We thank you for forwarding the most recent reviewer comments and for the opportunity to improve our article further. Below we detail how each point has been addressed in the accompanying revision.
We apologies for the oversight: during file conversion the new heading was inadvertently removed. We have now reinstated the subsection as
Missing Section 5.6
Action taken: We regret the omission. Section 5.6—now entitled “Candidozyma auris (Candida auris)”—has been re-inserted immediately after Section 5.5 (Other emerging yeast-like strains), highlighted in yellow for easy review.
Content: The subsection follows exactly the structure used for other species (epidemiology → resistance profile → clinical relevance) and incorporates the most recent regional data and references
All figures and tables have been repositioned to follow the paragraph in which they are first cited. For example, Figure 1 (Distribution of Candida spp.) now appears immediately after its first in-text mention instead of at the end of Section 2. Captions and in-text citations were checked to ensure consistency.
We believe these amendments fully resolve the reviewer’s remaining concerns and improve the manuscript’s clarity and coherence. We appreciate the reviewer’s careful reading.

Reviewer 3 Report

The authors have improved the manuscript in accordance with the comments. I propose to accept the revised manuscript for publication.

The authors have improved the manuscript in accordance with the comments. I propose to accept the revised manuscript for publication.

Author Response

Thank you very much for taking the time to re-evaluate our revised manuscript and for your positive recommendation to accept it for publication in the Journal of Fungi. We are delighted that our changes have addressed your concerns and helped improve the clarity and impact of our work. Your constructive feedback has been invaluable throughout this process.

Round 3

Reviewer 1 Report

The authors addressed my concerns.

The authors addressed my concerns.